# Diagnostic yield of nine user-friendly bioinformatics tools for predicting *Mycobacterium tuberculosis* drug resistance: A systematic review and network meta-analysis

Ya-Li Chen[1,2], Yu He[3], Victor Naestholt Dahl[4], Kan Yu[5,6], Yan-An Zhang[2,7], Cui-Ping Guan[1,2]*, Mao-Shui Wang[1,2]*

1 Department of Lab Medicine, Shandong Public Health Clinical Center, Shandong University, Jinan, China, 2 Shandong Key Laboratory of Infectious Respiratory Disease, Shandong Provincial Health Commission, Jinan, China, 3 Department of Clinical Laboratory, First Affiliated Hospital of Guangxi Medical University, Nanning, China, 4 Department of Infectious Diseases, Aarhus University Hospital, Aarhus, Denmark, 5 The Marshall Centre for Infectious Diseases, Research and Training, The University of Western Australia, Perth, Western Australia, Australia, 6 School of Biomedical Sciences, The University of Western Australia, Perth, Western Australia, Australia, 7 Department of Cardiovascular Surgery, Shandong Public Health Clinical Center, Shandong University, Jinan, China

* 317480883@qq.com (C-PG); wangmaoshui@gmail.com (M-SW)

## Abstract

To compare the diagnostic yield of various bioinformatics tools for predicting *Mycobacterium tuberculosis* drug resistance. A systematic review of PubMed, Embase, Scopus, Web of Science, CINAHL and the Cochrane Library was performed to identify studies reporting the effectiveness of bioinformatic tools for predicting resistance to anti-tuberculosis (TB) drugs. Data were collected and pooled using random-effects meta-analysis and Bayesian network meta-analysis (NMA). Summary receiver operating characteristic curves (SROCs) analysis were performed, and superiority index (SI) and area under the curve (AUC) were calculated. Thirty-three studies evaluated 9 different bioinformatics tools for predicting resistance to 14 anti-TB drugs. NMA and SROCs demonstrated that *TBProfiler*, *TGS-TB*, *Mykrobe*, *PhyResSE*, and *SAM-TB* all exhibited satisfactory performance. Remarkably, *TBProfiler* stood out with its exceptional ability to predict resistance to the majority of anti-TB drugs, including isoniazid (SI: 3.39 [95% confidence interval (CI): 0.20, 11.00]; AUC: 0.97 [0.95, 0.98]), rifampicin (SI: 6.38 [0.60, 15.00]; AUC: 0.99 [0.98, 1.00]), ethambutol (SI: 5.15 [0.60, 13.00]; AUC: 0.96 [0.94, 0.97]), streptomycin (SI: 3.67 [0.60, 11.00]; AUC: 0.97 [0.95, 0.98], amikacin (SI: 2.49 [0.14, 11.00]; AUC: 0.97 [0.96, 0.99]), kanamycin (SI: 2.26 [0.14, 9.00]; AUC: 0.98 [0.97, 0.99]), levofloxacin (SI: 1.87 [0.11, 9.00]; AUC: 0.95 [0.93, 0.97]), and prothionamide (SI: 2.73 [0.20, 7.00]; AUC: 0.87 [0.84, 0.90]). Meanwhile, *Mykrobe* demonstrated superior accuracy specifically for moxifloxacin (SI: 3.96 [0.11, 13.00]; AUC: 0.97 [0.95, 0.98]). Lastly, *TGS-TB* had the best efficacy in predicting resistance to pyrazinamide (SI: 12.53 [1.67, 17.00]; AUC: 0.97 [0.95, 0.98]), capreomycin (SI: 4.22 [0.08, 15.00]; AUC: 1.00 [0.98, 1.00]), and ethionamide (SI: 2.15 [0.33, 7.00];

**Data availability statement:** All relevant data for this study are publicly available. Key data are presented within the main text of this article. The complete dataset and associated analysis code are available as Supporting Information to this paper.

**Funding:** This work was supported by the Taishan Scholar Project of Shandong Province (Grant No. tsqn202211358 to MSW). The funders had no role in study design, data collection and analysis, decision to publish, or preparation of the manuscript.

**Competing interests:** The authors have declared that no competing interests exist.

AUC: 0.96 [0.94, 0.98]). *TBProfiler*, *TGS-TB*, *Mykrobe*, *PhyResSE* and *SAM-TB* have all demonstrated outstanding accuracy in predicting resistance to anti-TB drugs. In particular, *TBProfiler* stood out for its exceptional performance in predicting resistance to most anti-TB drugs, while *TGS-TB* excelled in predicting resistance to pyrazinamide and certain second-line drugs. The efficacy of *SAM-TB* requires further investigation to fully establish its reliability and effectiveness. To ensure the accuracy and reliability of genotypic drug susceptibility testing, bioinformatics tools should be refined and adapted continuously to accommodate novel and current resistance-associated mutations.

## Introduction

Drug-resistant tuberculosis (TB) remains a serious public health concern worldwide, complicating TB management and threatening global TB control [1,2]. Timely and precise identification of *Mycobacterium tuberculosis* (MTB) drug resistance is essential for improving patient outcomes and reducing the risk of worsening the current global situation. Until recently, culture-based phenotypic drug susceptibility testing (pDST) has been routinely used to detect drug-resistant MTB isolates. However, the pDST is time-consuming and requires specialized laboratory infrastructure, skilled personnel, and strict quality control protocols [3]. These drawbacks limit its clinical application, particularly in resource-constrained settings.

In recent decades, whole-genome sequencing (WGS) has been implemented for TB resistance profiling and has shown great potential. The sensitivity for detecting first-line drug resistance (e.g., isoniazid, rifampin, and ethambutol) has exceeded 90%, albeit considerably lower for pyrazinamide (51%) [4]. For second-line drug resistance, sensitivity has also shown promise, ranging from 85-96% for six second-line drugs (i.e., moxifloxacin, rifabutin, amikacin, kanamycin, streptomycin, and ethionamide) [4]. The WGS method has also shown superiority to pDST by identifying an additional 1% of likely drug-resistant isolates and offering a 20% improvement in the detection of resistance-conferring mutations compared to commercial genotypic assays [5]. To analyze WGS data, however, expertise and bioinformatics skills are needed. Several user-friendly bioinformatics tools, such as *TBprofiler* [6,7], *PhyResSE* [8], *KvarQ* [9], *Mykrobe* [10,11], and *TGS-TB* [12], have been developed to facilitate genomic profiling. Nevertheless, the diagnostic effectiveness of these tools varies widely. The sensitivity of *TBprofiler* ranges from 79-97%, that of *Mykrobe* ranges from 52-87%, and that of *PhyResSE ranges* from 57-90% [13,14]. Therefore, to help health professionals understand the diagnostic value of bioinformatics tools and aid in the clinical decision-making process, we aimed to compare the diagnostic accuracy of available bioinformatics tools for predicting MTB drug resistance via a systematic review.

## Materials and methods

This systematic review and network meta-analysis (NMA) was conducted in accordance with the Preferred Reporting Items for Systematic Review and Meta-analysis of Diagnostic Test Accuracy Studies and PRISMA for Network Meta-Analyses guidelines (Checklist, S1 File) [15,16]. The protocol was registered in PROSPERO (CRD42023470818).

### Search strategy

PubMed, Embase, Scopus, Web of Science, CINAHL and Cochrane Library databases were searched for articles published prior to October 25, 2024, without any filtering. The search

strategy is available in S2 File. The gray literature was searched using Google Scholar. However, no additional literature was identified.

## Eligibility

Studies evaluating the accuracy of bioinformatics tools for detecting MTB drug resistance were included and assessed if they met the following inclusion criteria: 1) written in English, 2) had at least one bioinformatics tool evaluated against culture-based pDST (recommended by the World Health Organization (WHO) [past and currently], or not [undefined]; S1 Table) [3,17–19], and 3) had a sample size ≥ 10 cases.

Duplicated studies, book chapters, conference abstracts, reviews, correspondences, notes, catalog-based approaches (wherein 1 causative mutation suggests resistance [e.g., the WHO catalog]; without a complicated algorithm), secondary analyses of publicly accessible sequence data, and studies without available data were excluded. All retrieved studies were independently evaluated by two reviewers (CYL and HY), and any discrepancies were discussed with a third reviewer (GCP).

## Data extraction

Data extraction was independently conducted by two reviewers (CYL and HY), and discrepancies were resolved through discussion. The variables included study characteristics (first author, publication year, and study period), strain characteristics (source of isolate, lineage, sample size, number of multidrug and extensively drug-resistant TB [M/XDR-TB]), sequencing methods (platform, quality, and data accession), pDST (DST method and critical concentrations of anti-TB drugs), bioinformatics tools (including version), and diagnostic yields such as true positive (TP), true negative (TN), false positive (FP), and false negative (FN) events compared to culture-based pDSTs.

## Bias assessment

The Quality Assessment of Diagnostic Accuracy Studies (QUADAS-2) was used for assessment of bias risk and applicability concerns [20]. In brief, the QUADAS-2 tool comprises four key domains: 1) patient selection, 2) index test, 3) reference standard, and 4) flow and timing. Each domain was evaluated by two independent reviewers (CYL and HY) for potential risk of bias, while concerns of applicability were assessed for the first three domains. The risk of each domain was rated as high, low, or unclear.

## Statistical analysis

Traditional meta-analyses were performed using Stata (Version 17, StataCorp LLC, College Station, TX). Bivariate random-effects meta-analysis was employed, and the *'metadta'* and *'midas'* packages were used to pool the diagnostic accuracy of the various bioinformatics tools compared against pDSTs. Pooled sensitivity and specificity were calculated, and a summary receiver operating characteristic (SROC) analysis was performed, reporting the area under the SROC curve. Additionally, between-study heterogeneity was quantified using the $I^2$ statistic [21]. Potential sources of heterogeneity were identified through subgroup analyses, such as categorization according to the WHO recommendations for pDSTs.

The Stata package *'metadta'* and help files, developed by Nyaga et al. [22,23], were publicly available [24]. The *midas* package was used to constructed a hierarchical SROC, including confidence and prediction regions [25]. By calculating the area under the curve (AUC), the diagnostic performance of each bioinformatics tool was categorized as excellent (≥ 0.97), very

good (0.93-0.96), good (0.75-0.92), or fair (0.50-0.75) [26]. Publication bias was evaluated by Deek's funnel plot [27,28]. A *p* value less than 0.05 was considered to indicate statistical significance.

Bayesian NMAs were performed and visualized by a graph comprising nodes (different bioinformatics tools) and edges (head-to-head comparisons). The node size represented the number of studies using the different bioinformatics tools, while thicker edges reflected a greater frequency of comparisons between the tools. To integrate the results of different studies in which multiple tools were compared, Bayesian NMAs were conducted using an ANOVA model, which comprised three chains of 15,000 iterations with 5,000 iterations of burn-in and a thinning interval of five. The model was implemented within a Bayesian framework using Stan software, executed via the '*rstan*' package in R (using streptomycin as an example; S3 File; All origianal data for analysis, S2 Table). The Stan code for this model is kindly provided by Nyaga et al. in their Supplementary Material [29]. 'Dis' signified the count of patients confirmed with the disease based on the gold standard diagnostic method, whereas 'NDis' denoted the count of individuals excluded from having the disease according to the same gold standard (S3 File). According to the following sequence, *TBProfiler*, *Mykrobe*, *PhyResSE*, *TGS-TB*, *KvarQ*, *CASTB*, *MTBseq*, *SAM-TB*, and *GenTB* were assigned as tests 1–9. Relative sensitivities and specificities and superiority indices (SIs) were used to describe each bioinformatics tool [22,23]. Specifically, the SI was used to quantify the probability rank of bioinformatics tools, where a higher value indicates greater predictive accuracy [30].

## Results

### Literature selection

A total of 1325 records were identified through database searching (Fig 1). Of these, 680 records were duplicates (n=491) or removed for other reasons (n=189: non-English, n=12; book chapter, n=28; conference abstract, n=46; review, n=88; correspondence or note, n=15). After screening the studies, 553 additional records were excluded. Finally, 33 articles were eligible for analysis (S4 File).

### Study characteristics and risk of bias

All included studies were published after 2016, and the study period ranged from 2 months to 15 years. A total of 8010 strains were analyzed, approximately half of which (31.4%, n= 2516) were identified as M/XDR-TB. *TBProfiler* (n=23), *Mykrobe* (n=16) and *PhyResSE* (n=12) were the three most frequently evaluated tools. The study sample size varied between 10 and 3808. Seventeen studies were conducted in Asia, including China (n=7), Indonesia (n=2), Korea (n=2), Thailand (n=2), Japan (n=1), Kazakhstan (n=1), India (n=1), and Singapore (n=1). The raw sequencing data were available from 23 studies. The Illumina sequencing platform (Illumina, Inc., CA, USA) was used in most studies (n=30) for genomic sequencing, with one study concurrently using the Nanopore platform (Oxford Nanopore Technologies PLC, Oxford, UK). Most studies (n=32) used the proportion method for pDSTs on conventional solid and liquid media, such as the Löwenstein–Jensen (LJ), Middlebrook 7H10/7H11 and BACTEC MGIT 960 systems. The MGIT 960 or pyrazinamidase activity assays were predominantly employed for pyrazinamide resistance testing. Nine different bioinformatics tools (*MTBseq*, *PhyResSE*, *KvarQ*, *TBProfiler*, *CASTB*, *Mykrobe*, *SAM-TB*, GenTB and *TGS-TB*) were evaluated for predicting MTB resistance to first- (isoniazid, rifampicin, pyrazinamide, ethambutol, and streptomycin) and second-line (amikacin, capreomycin, ethionamide, prothionamide, kanamycin, levofloxacin, moxifloxacin, ofloxacin, and para-aminosalicylic acid)

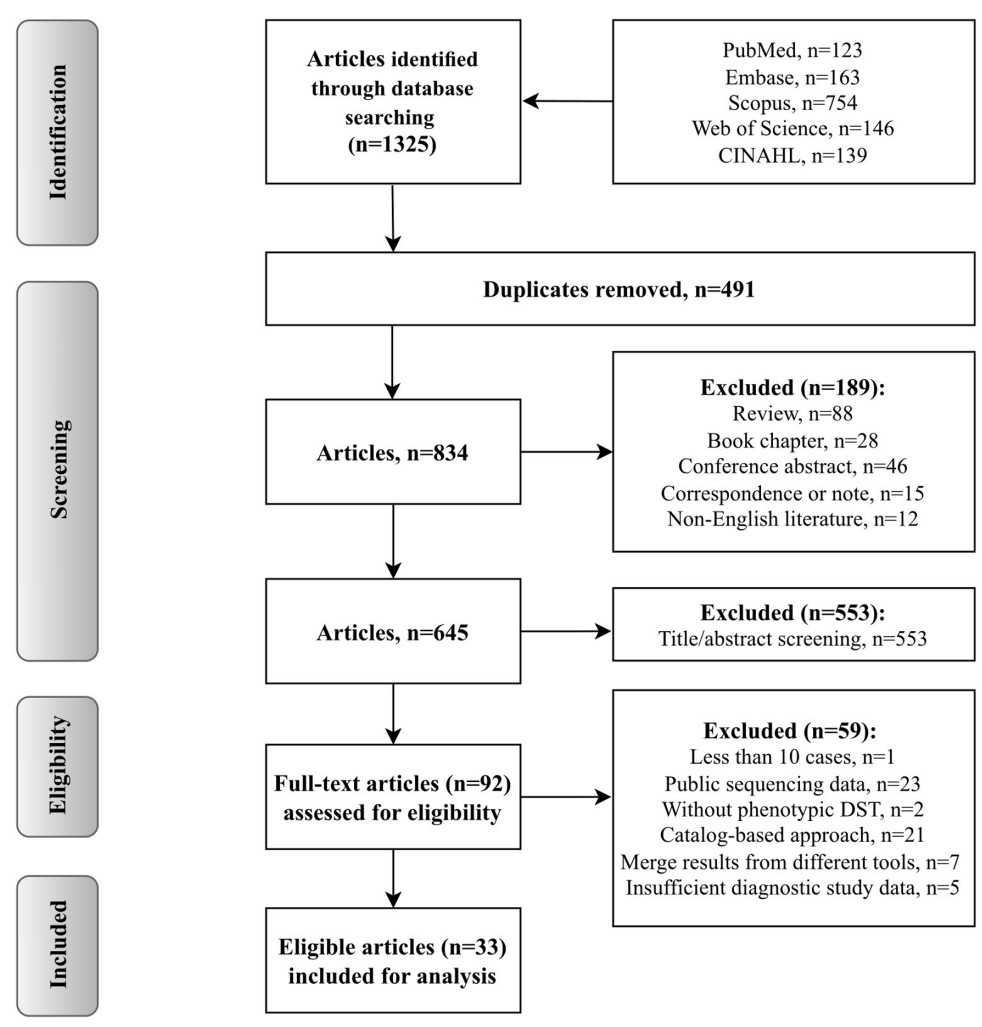

**Fig 1. Flow chart of study selection.**

anti-TB drugs. The features of the bioinformatics tools used are presented in S3 Table, while the characteristics of the included studies can be found in S1 Table.

Regarding sample and patient selection, most studies had a high risk of bias (n=25/33, 76%), and significant concerns of applicability (n=19/33, 58%), were explained by the unblinded MTB resistance profiling. For the index and reference test domains, the risk of bias of all studies was evaluated as low or low applicability. Regarding the domain of flow and timing, low bias risk was also considered (S1 Fig and S4 Table).

## Accuracy of bioinformatics tools for predicting MTB drug resistance

**First-line drugs.** The predictive role of bioinformatics tools for MTB resistance profiling is detailed in Table 1 and S2 and S3 Figs. *TBProfiler* demonstrated excellent performance in detecting resistance to isoniazid (AUC, 0.97 [0.95, 0.98]), rifampicin (AUC, 0.99 [0.98, 1.00]), and streptomycin (AUC, 0.97 [0.95, 0.98]), and very good performance for ethambutol (AUC, 0.96 [0.94, 0.97]) and pyrazinamide (AUC, 0.94 [0.92, 0.96]). *Mykrobe* showed excellent performance for rifampicin (AUC, 0.99 [0.97, 0.99]) and very good performance for isoniazid (AUC, 0.95 [0.93, 0.97]), pyrazinamide (AUC, 0.94 [0.91, 0.96]), and streptomycin (AUC,

**Table 1. Diagnostic accuracy of bioninformatics tools in predicting resistance to all the first-line anti-TB drugs.**

| Parameter | TBProfiler | Mykrobe | PhyResSE | TGS-TB | KvarQ | CASTB |
|---|---|---|---|---|---|---|
| Isoniazid | | | | | | |
| Sensitivity (95%CI) | 0.94 (0.91-0.96) | 0.88 (0.80-0.93) | 0.96 (0.81-0.99) | 0.87 (0.65-0.96) | 0.85 (0.75-0.92) | 0.78 (0.63-0.88) |
| Specificity (95%CI) | 0.95 (0.89-0.98) | 0.97 (0.85-0.99) | 0.99 (0.81-1.00) | 0.76 (0.34-0.95) | 0.99 (0.95-1.00) | 0.88 (0.42-0.99) |
| AUC (95%CI) | 0.97 (0.95-0.98) | 0.95 (0.93-0.97) | 0.99 (0.98-1.00) | 0.90 (0.87-0.92) | 0.96 (0.94-0.98) | 0.84 (0.81-0.87) |
| $I^2$ (%)-Sensitivity | 31.43 | 60.64 | 38.11 | 62.88 | 32.85 | 40.49 |
| $I^2$ (%)-Specificity | 25.28 | 28.10 | 0.34 | 88.40 | 22.14 | 70.18 |
| Rifampicin | | | | | | |
| Sensitivity (95%CI) | 0.98 (0.96-0.99) | 0.94 (0.89-0.97) | 0.99 (0.69-1.00) | 0.97 (0.76-1.00) | 0.91 (0.83-0.95) | 0.90 (0.57-0.98) |
| Specificity (95%CI) | 0.97 (0.92-0.99) | 0.97 (0.94-0.98) | 0.99 (0.88-1.00) | 0.81 (0.00-1.00) | 0.99 (0.85-1.00) | 0.92 (0.84-0.96) |
| AUC (95%CI) | 0.99 (0.98-1.00) | 0.99 (0.97-0.99) | 1.00 (0.99-1.00) | 0.98 (0.96-0.99) | 0.93 (0.91-0.95) | 0.96 (0.94-0.97) |
| $I^2$ (%)-Sensitivity | 9.07 | 27.52 | 0.02 | 19.33 | 2.38 | 44.74 |
| $I^2$ (%)-Specificity | 35.90 | 36.09 | 2.68 | 0.00 | 26.58 | 34.49 |
| Ethambutol | | | | | | |
| Sensitivity (95%CI) | 0.92 (0.87-0.95) | 0.72 (0.64-0.80) | 0.84 (0.72-0.91) | 0.95 (0.81-0.99) | 0.64 (0.42-0.82) | 0.75 (0.45-0.92) |
| Specificity (95%CI) | 0.88 (0.80-0.93) | 0.92 (0.82-0.97) | 0.94 (0.87-0.97) | 0.87 (0.71-0.95) | 0.97 (0.94-0.99) | 0.84 (0.07-1.00) |
| AUC (95%CI) | 0.96 (0.94-0.97) | 0.83 (0.80-0.86) | 0.86 (0.83-0.89) | 0.97 (0.95-0.98) | 0.95 (0.93-0.97) | 0.80 (0.76-0.83) |
| $I^2$ (%)-Sensitivity | 16.10 | 14.60 | 0.32 | 7.00 | 14.76 | 33.35 |
| $I^2$ (%)-Specificity | 54.99 | 37.51 | 72.99 | 75.81 | 33.17 | 1.70 |
| Pyrazinamide | | | | | | |
| Sensitivity (95%CI) | 0.78 (0.67-0.86) | 0.85 (0.72-0.92) | 0.84 (0.59-0.95) | 0.96 (0.92-0.98) | 0.44 (0.20-0.71) | 0.33 (0.24-0.44) |
| Specificity (95%CI) | 0.98 (0.95-1.00) | 0.95 (0.82-0.99) | 0.98 (0.91-0.99) | 0.99 (0.95-1.00) | 0.96 (0.90-0.98) | 0.98 (0.93-0.99) |
| AUC (95%CI) | 0.94 (0.92-0.96) | 0.94 (0.91-0.96) | 0.98 (0.96-0.99) | 0.97 (0.95-0.98) | 0.89 (0.86-0.91) | 0.42 (0.38-0.46) |
| $I^2$ (%)-Sensitivity | 40.02 | 32.23 | 55.90 | 0.04 | 47.62 | 0.39 |
| $I^2$ (%)-Specificity | 8.47 | 10.00 | 49.94 | 29.88 | 57.44 | 26.59 |
| Streptomycin | | | | | | |
| Sensitivity (95%CI) | 0.92 (0.86-0.95) | 0.81 (0.64-0.91) | 0.85 (0.73-0.93) | 0.87 (0.70-0.95) | 0.67 (0.48-0.81) | 0.62 (0.29-0.87) |
| Specificity (95%CI) | 0.95 (0.86-0.98) | 0.97 (0.81-1.00) | 0.98 (0.92-1.00) | 0.95 (0.87-0.98) | 1.00 (0.97-1.00) | 0.94 (0.00-1.00) |
| AUC (95%CI) | 0.97 (0.95-0.98) | 0.94 (0.91-0.96) | 0.97 (0.95-0.98) | 0.97 (0.95-0.98) | 0.97 (0.96-0.99) | 0.69 (0.65-0.73) |
| $I^2$ (%)-Sensitivity | 43.26 | 54.85 | 58.00 | 51.76 | 31.46 | 55.55 |
| $I^2$ (%)-Specificity | 18.01 | 2.25 | 17.68 | 33.70 | 1.87 | 0.00 |

Abbreviations: CI; confidence interval. AUC, Area under the ROC curve.

0.94 [0.91, 0.96]). *PhyResSE* exhibited excellent performance for isoniazid (AUC, 0.99 [0.98, 1.00]), rifampicin (AUC, 1.00 [0.99, 1.00]), pyrazinamide (AUC, 0.98 [0.96, 0.99]), and streptomycin (AUC, 0.97 [0.95, 0.98]). *TGS-TB* performed excellently for rifampicin (AUC, 0.98 [0.96, 0.99]), ethambutol (AUC, 0.97 [0.95, 0.98]), pyrazinamide (AUC, 0.97 [0.95, 0.98]), and streptomycin (AUC, 0.97 [0.95, 0.98]). *KvarQ* was excellent for streptomycin (AUC, 0.97 [0.96, 0.99]) and very good for isoniazid (AUC, 0.96 [0.94, 0.98]), rifampicin (AUC, 0.93 [0.91, 0.95]), and ethambutol (AUC, 0.95 [0.93, 0.97]). *CASTB* performed very good solely for rifampicin (AUC, 0.96 [0.94, 0.97]).

**Second-line drugs.** When evaluating injectable aminoglycoside antibiotics such as amikacin, capreomycin, and kanamycin, both *TBProfiler* (AUC, 0.97 [0.96, 0.99]; AUC, 1.00 [0.98, 1.00]; AUC, 0.98 [0.97, 0.99]) and *PhyResSE* (AUC, 0.99 [0.98, 1.00]; AUC, 1.00 [0.99, 1.00]; AUC, 0.98 [0.96, 0.99]) demonstrated excellent performance, while *Mykrobe* (AUC, 1.00 [0.98, 1.00]; AUC, 0.99 [0.97, 0.99]; AUC, 0.96 [0.94, 0.97]) and *TGS-TB* (AUC, 0.96 [0.94, 0.97]; AUC, 1.00 [0.98, 1.00]; AUC, 1.00 [0.99, 1.00]) also showed high efficacy. For

fluoroquinolones, including levofloxacin, moxifloxacin, and ofloxacin, *TBProfiler* (AUC, 0.95 [0.93, 0.97]; AUC, 0.98 [0.96, 0.99]; AUC, 0.94 [0.92, 0.96]) exhibited very good performance. Notably, *Mykrobe* demonstrated excellent performance specifically for moxifloxacin (AUC, 0.97 [0.95, 0.98]) and ofloxacin (AUC, 0.98 [0.96, 0.99]). Furthermore, *TGS-TB* displayed very good performance in assessing ethionamide (AUC, 0.96 [0.94, 0.98]). Additional details on the predictive values are shown in Table 2 and S2 and S3 Figs.

## Subgroup analyses

Subgroup analyses stratified by pDST revealed that compared with the current WHO recommendations, *TBProfiler* had a lower sensitivity for predicting kanamycin resistance (sensitivity: 0.82 [0.73, 0.94] vs. 1.00 [0.00, 1.00]; *P*<0.01) (S5 Table and S2 Fig).

Deek's funnel plots revealed no evidence of publication bias across the studies, except for specific analyses (illustrated in S4 Fig): *Mykrobe* for capreomycin and streptomycin resistance, *TBProfiler* for isoniazid and pyrazinamide resistance, and *PhyResSE* for ethionamide resistance.

## Bayesian network meta-analyses

The SI, relative sensitivity, and relative specificity are detailed in Tables 3, 4 and S6, respectively. The network plots of all the bioinformatics tools are displayed in S5 Fig.

**First-line drugs.** For isoniazid, *SAM-TB* (SI, 4.16 [0.08, 13.00]) showed the highest SI, followed by *KvarQ* (SI, 3.73 [0.11, 11.00]), *TBProfiler* (SI, 3.39 [0.20, 13.00]), and *PhyResSE* (SI, 2.92 [0.14, 9.05]). For rifampicin, *SAM-TB* (SI, 6.84 [0.09, 17.00]) had the highest SI, with *TBProfiler* (SI, 6.38 [0.60, 15.00]), *Mykrobe* (SI, 2.96 [0.11, 11.00]), and *PhyResSE* (SI, 2.51 [0.10, 11.00]) following. For ethambutol, *TBProfiler* (SI, 5.15 [0.60, 13.00]) topped with the highest SI, then *TGS-TB* (SI, 3.60 [0.14, 13.00]), *PhyResSE* (SI, 2.61 [0.14, 9.00]), and *KvarQ* (SI, 2.52 [0.20, 9.00]). For pyrazinamide, *TGS-TB* (SI, 12.53 [1.67, 17.00]) led with the highest SI, with *TBProfiler* (SI, 4.84 [0.60, 13.00]), *SAM-TB* (SI, 3.67 [0.14, 13.00]) and *Mykrobe* (SI, 2.68 [0.20, 11.00]) ranking next. For streptomycin, *SAM-TB* (SI, 5.96 [0.14, 15.00]) had the highest SI, followed by *KvarQ* (SI, 4.52 [0.20, 13.00]), *TBProfiler* (SI, 3.67 [0.60, 11.00]), and *PhyResSE* (SI, 2.83 [0.27, 11.00]). Of note is that if a tool was evaluated only in a single study, it was excluded from the conclusion despite having a high SI ranking. Likewise, SAM-TB, which was only assessed in two studies, should be interpreted cautiously.

**Second-line drugs.** For the aminoglycosides amikacin and kanamycin, *SAM-TB* (SI, 4.30 [0.08, 15.00]; SI, 6.61 [0.14, 17.00]) exhibited the highest SI, whereas for capreomycin, *TGS-TB* (SI, 4.22 [0.08, 15.00]) led the rankings, with *PhyResSE*, *Mykrobe*, and *TBProfiler* trailing closely behind. Regarding fluoroquinolones, *TBProfiler* (SI, 1.87 [0.11, 9.00]) ranked highest for levofloxacin, *Mykrobe* (SI, 3.96 [0.11, 13.00]) for moxifloxacin, and *TGS-TB* (SI, 3.19 [0.14, 9.00]) for ofloxacin. Additionally, *TGS-TB* (SI, 2.15 [0.33, 7.00]) topped the list for ethionamide, *TBProfiler* (SI, 2.73 [0.20, 7.00]) for prothionamide, and *Mykrobe* (SI, 4.67 [0.33, 11.00]) for para-aminosalicylic acid. For this analysis, SI rankings from tools only used in a single study was also ignored.

## Discussion

Currently, several bioinformatics tools have been developed to process WGS data for MTB drug resistance prediction, showing promising potential for tailoring anti-TB regimens. To date, limited information on comparisons between different bioinformatics tools are available. Few studies have been conducted evaluating different tools on the same samples have been conducted.

**Table 2. Diagnostic accuracy of bioinformatics tools in predicting resistance to any second-line anti-TB drugs.**

| Parameter | TBProfiler | Mykrobe | PhyResSE | TGS-TB |
|---|---|---|---|---|
| Amikacin | | | | |
| Sensitivity (95%CI) | 0.90 (0.74-0.96) | 0.90 (0.62-0.98) | 0.97 (0.29-1.00) | 0.79 (0.47-0.94) |
| Specificity (95%CI) | 0.99 (0.95-1.00) | 0.98 (0.95-0.99) | 0.98 (0.92-1.00) | 1.00 (0.90-1.00) |
| AUC (95%CI) | 0.97 (0.96-0.99) | 1.00 (0.98-1.00) | 0.99 (0.98-1.00) | 0.96 (0.94-0.97) |
| $I^2$ (%)-Sensitivity | 18.53 | 20.24 | 3.88 | 9.86 |
| $I^2$ (%)-Specificity | 10.17 | 3.88 | 31.53 | 8.03 |
| Capreomycin | | | | |
| Sensitivity (95%CI) | 0.94 (0.85-0.98) | 0.73 (0.24-0.96) | 0.94 (0.68-0.99) | 0.93 (0.50-0.99) |
| Specificity (95%CI) | 0.99 (0.98-0.99) | 0.98 (0.96-0.99) | 0.99 (0.96-1.00) | 0.99 (0.95-1.00) |
| AUC (95%CI) | 1.00 (0.98-1.00) | 0.99 (0.97-0.99) | 1.00 (0.99-1.00) | 1.00 (0.98-1.00) |
| $I^2$ (%)-Sensitivity | 0.00 | 16.34 | 0.00 | 0.31 |
| $I^2$ (%)-Specificity | 0.00 | 1.18 | 0.00 | 0.25 |
| Kanamycin | | | | |
| Sensitivity (95%CI) | 0.94 (0.77-0.98) | 0.69 (0.42-0.87) | 0.79 (0.29-0.97) | 0.98 (0.18-1.00) |
| Specificity (95%CI) | 0.98 (0.88-1.00) | 0.98 (0.91-1.00) | 0.98 (0.87-1.00) | 0.99 (0.79-1.00) |
| AUC (95%CI) | 0.98 (0.97-0.99) | 0.96 (0.94-0.97) | 0.98 (0.96-0.99) | 1.00 (0.99-1.00) |
| $I^2$ (%)-Sensitivity | 18.40 | 13.05 | 30.67 | 3.25 |
| $I^2$ (%)-Specificity | 4.02 | 10.80 | 48.74 | 22.78 |
| Levofloxacin | | | | |
| Sensitivity (95%CI) | 0.93 (0.45-0.99) | – | 0.58 (0.49-0.67) | – |
| Specificity (95%CI) | 0.95 (0.92-0.97) | – | 0.96 (0.92-0.98) | – |
| AUC (95%CI) | 0.95 (0.93-0.97) | – | 0.92 (0.89-0.94) | – |
| $I^2$ (%)-Sensitivity | 16.87 | – | 0.00 | – |
| $I^2$ (%)-Specificity | 0.00 | – | 0.00 | – |
| Moxifloxacin | | | | |
| Sensitivity (95%CI) | 0.88 (0.69-0.96) | 0.77 (0.33-0.96) | 0.50 (0.40-0.60) | – |
| Specificity (95%CI) | 0.97 (0.89-0.99) | 0.98 (0.54-1.00) | 0.92 (0.78-0.97) | – |
| AUC (95%CI) | 0.98 (0.96-0.99) | 0.97 (0.95-0.98) | 0.72 (0.68-0.76) | – |
| $I^2$ (%)-Sensitivity | 41.11 | 33.17 | 0.00 | – |
| $I^2$ (%)-Specificity | 27.45 | 7.76 | 1.27 | – |
| Ofloxacin | | | | |
| Sensitivity (95%CI) | 0.92 (0.89-0.94) | 0.99 (0.18-1.00) | – | – |
| Specificity (95%CI) | 0.97 (0.93-0.98) | 0.96 (0.91-0.98) | – | – |
| AUC (95%CI) | 0.94 (0.92-0.96) | 0.98 (0.96-0.99) | – | – |
| $I^2$ (%)-Sensitivity | 0.92 | 1.44 | – | – |
| $I^2$ (%)-Specificity | 19.95 | 4.80 | – | – |
| Para-aminosalicylic acid | | | | |
| Sensitivity (95%CI) | 0.68 (0.57-0.78) | – | – | – |
| Specificity (95%CI) | 0.98 (0.94-0.99) | – | – | – |
| AUC (95%CI) | 0.80 (0.76-0.83) | – | – | – |
| $I^2$ (%)-Sensitivity | 2.11 | – | – | – |
| $I^2$ (%)-Specificity | 28.58 | – | – | – |
| Ethionamide | | | | |
| Sensitivity (95%CI) | 0.85 (0.75-0.92) | – | 0.68 (0.35-0.89) | 0.95 (0.79-0.99) |
| Specificity (95%CI) | 0.80 (0.67-0.88) | – | 0.85 (0.58-0.96) | 0.83 (0.51-0.96) |
| AUC (95%CI) | 0.90 (0.87-0.92) | – | 0.80 (0.76-0.83) | 0.96 (0.94-0.98) |
| $I^2$ (%)-Sensitivity | 20.70 | – | 7.51 | 17.16 |

*(Continued)*

**Table 2.** (Continued)

| Parameter | TBProfiler | Mykrobe | PhyResSE | TGS-TB |
|---|---|---|---|---|
| I² (%)-Specificity | 57.90 | – | 71.86 | 78.49 |
| Prothionamide | | | | |
| Sensitivity (95%CI) | 0.89 (0.43-0.99) | – | – | – |
| Specificity (95%CI) | 0.85 (0.79-0.90) | – | – | – |
| AUC (95%CI) | 0.87 (0.84-0.90) | – | – | – |
| I² (%)-Sensitivity | 50.30 | – | – | – |
| I² (%)-Specificity | 5.85 | – | – | – |

Abbreviations: CI, confidence interval. AUC, Area under the ROC curve.

Our review and meta-analyses revealed that *TBProfiler*, *TGS-TB*, *Mykrobe,* and *PhyResSE* exhibit exceptional accuracy in predicting resistance to the majority of anti-TB drugs, with *TBProfiler* being particularly excellent. Further NMA analyses confirmed the meta-analysis results, also finding that *SAM-TB* had a good performance. Remarkably, according to the SIs, the ranking performance of each bioinformatics tool varied depending on the specific anti-TB drug being tested. This means that each bioinformatics tool may be indicated for specific anti-TB drugs, with sequence mutations not being the only source for constructing these bioinformatics tools. Additionally, running different tools in parallel could offer a more robust approach for examining drug resistance.

The catalog of MTB mutations issued by the WHO serves as a valuable reference for comparing sequencing results when interpreting resistance in MTB isolates. Usually, statistical algorithms and expert-derived rules are used to grade potential genetic mutations associated with DST resistance. In 2023, the catalog was updated, leading to a significant improvement in resistance detection for new and repurposed drugs, including bedaquiline, delamanid, linezolid, and clofazimine, although their accuracy remained less than 50% [31]. The data suggest that >80% of the genes included in the catalog are sensitive to most drugs (such as rifampicin, isoniazid, ethambutol, fluoroquinolones, and streptomycin), but this percentage is lower for ethionamide (75.7%). On the other hand, the specificity has been reported to be >95% for rifampicin, isoniazid, fluoroquinolones, and streptomycin but slightly lower for ethionamide (91.4%), moxifloxacin (91.6%), and ethambutol (93.3%) [19,32].

Previously, a mutational catalog-based approach was thought to be comparable or equal to that of bioinformatics tools [33]. However, the latter (i.e., machine learning) outperforms the former (catalog-based method), due to its powerful, cost-effective, and user-friendly design, enabling clinicians with minimal or no experience in WGS analysis to swiftly conduct comprehensive drug resistance analysis [34]. Still, inexperienced personnel should perform these analyses with caution, if unsupervised, even with user-friendly and high-performing tools, due to the potential clinical implications. Due to the excellent performance for most anti-TB drugs, our findings suggest that *TBProfiler* should be considered the most useful tool for the prediction of TB resistance with a sensitivity of >80% and specificity of >95% for predicting first-line drug resistance, with the exceptions of ethambutol and pyrazinamide resistance. In the case of pyrazinamide resistance, *TBProfiler* exhibits a relatively low sensitivity of 78%, whereas *TGS-TB* excels with a pooled sensitivity of 96%, specificity of 99% and an exceptionally high SI of 12.53 [1.67, 17.00], outperforming other tools due to its advantages in terms of insertion and deletion detection [2,35]. For ethambutol resistance, however, we found that *TBProfiler* has a relatively low specificity of 88%, which may be explained by the variability of pDSTs [36], borderline resistance [37], and heteroresistance [38]. For ethionamide, the

**Table 3. Bayesian network meta-analysis of the diagnostic accuracy of bioinformatics tools for predicting resistance to all the first-line anti-TB drugs.**

| Tools | Studies (M/XDR; sizes) | Absolute Sensitivity (95% CI) | Absolute Specificity (95% CI) | Superiority Index | Relative Sensitivity (95% CI) | Relative Specificity (95% CI) |
|---|---|---|---|---|---|---|
| Isoniazid | | | | | | |
| TBProfiler | 22 (1494; 6259) | 0.93 (0.91-0.95) | 0.88 (0.81-0.93) | 3.39 (0.20-11.00) | 0.99 (0.92-1.37) | 1.15 (0.92-1.83) |
| Mykrobe | 14 (312; 1274) | 0.90 (0.86-0.93) | 0.91 (0.83-0.95) | 1.94 (0.11-9.00) | 0.96 (0.87-1.32) | 1.18 (0.96-1.86) |
| PhyResSE | 10 (459; 1429) | 0.90 (0.86-0.94) | 0.92 (0.86-0.96) | 2.92 (0.14-9.05) | 0.96 (0.87-1.34) | 1.20 (0.97-1.89) |
| TGS-TB | 4 (212; 502) | 0.91 (0.84-0.96) | 0.74 (0.53-0.89) | 0.35 (0.06-3.00) | 0.97 (0.85-1.31) | 0.96 (0.65-1.54) |
| KvarQ | 4 (76; 461) | 0.89 (0.83-0.94) | 0.94 (0.85-0.99) | 3.73 (0.11-11.00) | 0.95 (0.84-1.29) | 1.22 (0.97-1.92) |
| CASTB | 4 (40; 209) | 0.90 (0.83-0.94) | 0.90 (0.77-0.96) | 1.89 (0.08-9.00) | 0.95 (0.84-1.30) | 1.16 (0.91-1.81) |
| MTBseq | 2 (621; 748) | 0.96 (0.91-0.99) | 0.72 (0.46-0.91) | 1.78 (0.11-9.00) | 1.03 (0.92-1.42) | 0.93 (0.56-1.49) |
| SAM-TB | 2 (152; 198) | 0.93 (0.83-0.98) | 0.87 (0.69-0.96) | 4.16 (0.08-13.00) | 0.99 (0.85-1.33) | 1.13 (0.86-1.71) |
| GenTB | 1 (100; 110) | 0.95 (0.68-1.00) | 0.80 (0.48-0.95) | 4.08 (0.07-15.00) | 1.00 (1.00-1.00) | 1.00 (1.00-1.00) |
| Rifampicin | | | | | | |
| TBProfiler | 22 (1494; 6259) | 0.96 (0.92-0.98) | 0.89 (0.79-0.96) | 6.38 (0.60-15.00) | 1.11 (0.93-2.06) | 1.25 (0.95-2.30) |
| Mykrobe | 14 (312; 1274) | 0.93 (0.85-0.97) | 0.88 (0.77-0.96) | 2.96 (0.11-11.00) | 1.08 (0.87-1.97) | 1.24 (0.94-2.26) |
| PhyResSE | 10 (459; 1429) | 0.88 (0.77-0.95) | 0.91 (0.81-0.97) | 2.51 (0.11-11.00) | 1.02 (0.81-1.88) | 1.28 (0.98-2.37) |
| TGS-TB | 4 (212; 502) | 0.88 (0.69-0.98) | 0.83 (0.58-0.95) | 1.33 (0.07-11.00) | 1.02 (0.72-1.93) | 1.15 (0.75-2.04) |
| KvarQ | 4 (76; 461) | 0.87 (0.69-0.97) | 0.89 (0.75-0.98) | 2.29 (0.08-13.00) | 1.00 (0.71-1.84) | 1.25 (0.93-2.26) |
| CASTB | 4 (40; 209) | 0.87 (0.67-0.97) | 0.87 (0.72-0.96) | 1.64 (0.08-11.00) | 1.01 (0.71-1.83) | 1.22 (0.89-2.19) |
| MTBseq | 2 (621; 748) | 0.90 (0.68-0.99) | 0.77 (0.47-0.93) | 1.33 (0.06-11.00) | 1.04 (0.71-1.93) | 1.08 (0.63-1.83) |
| SAM-TB | 2 (152; 198) | 0.91 (0.70-1.00) | 0.90 (0.73-0.99) | 6.84 (0.09-17.00) | 1.06 (0.74-2.00) | 1.27 (0.94-2.22) |
| GenTB | 1 (100; 110) | 0.91 (0.46-1.00) | 0.75 (0.39-0.94) | 2.07 (0.06-13.00) | 1.00 (1.00-1.00) | 1.00 (1.00-1.00) |
| Ethambutol | | | | | | |
| TBProfiler | 22 (1494; 6259) | 0.91 (0.87-0.94) | 0.82 (0.78-0.85) | 5.15 (0.60-13.00) | 1.12 (0.96-1.47) | **1.30 (1.07-1.65)** |
| Mykrobe | 14 (312; 1274) | 0.73 (0.64-0.82) | 0.85 (0.79-0.89) | 1.48 (0.11-7.00) | 0.90 (0.74-1.19) | **1.34 (1.11-1.70)** |
| PhyResSE | 10 (459; 1429) | 0.86 (0.78-0.92) | 0.82 (0.77-0.87) | 2.61 (0.14-9.00) | 1.05 (0.89-1.38) | **1.30 (1.08-1.64)** |
| TGS-TB | 4 (212; 502) | 0.88 (0.77-0.96) | 0.81 (0.73-0.88) | 3.60 (0.14-13.00) | 1.09 (0.89-1.42) | **1.28 (1.03-1.63)** |
| KvarQ | 4 (76; 461) | 0.66 (0.47-0.83) | 0.89 (0.83-0.94) | 2.52 (0.20-9.00) | 0.81 (0.54-1.15) | **1.41 (1.16-1.80)** |
| CASTB | 4 (40; 209) | 0.68 (0.47-0.85) | 0.84 (0.76-0.91) | 1.21 (0.07-7.00) | 0.84 (0.57-1.17) | **1.33 (1.08-1.71)** |
| MTBseq | 1 (16; 71) | 0.82 (0.45-0.99) | 0.84 (0.71-0.93) | 5.65 (0.08-17.00) | 1.01 (0.53-1.43) | **1.33 (1.05-1.73)** |
| SAM-TB | 2 (152; 198) | 0.80 (0.64-0.91) | 0.82 (0.73-0.89) | 1.91 (0.09-11.00) | 0.98 (0.76-1.28) | **1.30 (1.07-1.63)** |
| GenTB | 1 (100; 110) | 0.82 (0.62-0.94) | 0.64 (0.49-0.77) | 0.21 (0.06-1.00) | 1.00 (1.00-1.00) | 1.00 (1.00-1.00) |
| Pyrazinamide | | | | | | |
| TBProfiler | 15 (819; 5336) | 0.74 (0.62-0.83) | 0.94 (0.88-0.97) | 4.84 (0.60-13.00) | 0.91 (0.69-1.36) | **1.07 (1.01-1.17)** |
| Mykrobe | 9 (260; 641) | 0.69 (0.52-0.83) | 0.93 (0.86-0.97) | 2.68 (0.20-11.00) | 0.84 (0.60-1.30) | **1.06 (1.01-1.16)** |
| PhyResSE | 11 (478; 1436) | 0.70 (0.56-0.81) | 0.89 (0.81-0.94) | 0.34 (0.08-1.00) | 0.86 (0.62-1.31) | 1.01 (0.96-1.10) |
| TGS-TB | 5 (377; 693) | 0.91 (0.77-0.98) | 0.94 (0.86-0.98) | 12.53 (1.67-17.00) | 1.12 (0.85-1.70) | 1.07 (1.00-1.19) |
| KvarQ | 4 (76; 461) | 0.46 (0.22-0.71) | 0.91 (0.82-0.96) | 0.28 (0.07-1.00) | 0.56 (0.26-1.00) | 1.03 (0.96-1.14) |
| CASTB | 4 (195; 390) | 0.47 (0.20-0.72) | 0.93 (0.86-0.98) | 1.29 (0.09-7.00) | 0.57 (0.23-1.01) | 1.07 (1.00-1.16) |
| MTBseq | 1 (16; 71) | 0.49 (0.08-0.91) | 0.90 (0.78-0.97) | 0.87 (0.06-7.00) | 0.60 (0.09-1.28) | 1.02 (0.90-1.14) |
| SAM-TB | 2 (152; 198) | 0.85 (0.57-0.98) | 0.91 (0.83-0.96) | 3.67 (0.14-13.00) | 1.03 (0.65-1.59) | 1.04 (0.97-1.13) |
| GenTB | 2 (100; 144) | 0.84 (0.54-0.98) | 0.88 (0.78-0.94) | 1.33 (0.08-7.00) | 1.00 (1.00-1.00) | 1.00 (1.00-1.00) |
| Streptomycin | | | | | | |
| TBProfiler | 20 (1432; 1047) | 0.89 (0.84-0.92) | 0.83 (0.74-0.89) | 3.67 (0.60-11.00) | 0.94 (0.85-1.22) | 1.04 (0.83-1.57) |
| Mykrobe | 13 (308; 209) | 0.79 (0.70-0.86) | 0.86 (0.75-0.94) | 1.49 (0.14-7.00) | 0.83 (0.71-1.07) | 1.08 (0.85-1.66) |
| PhyResSE | 8 (451; 461) | 0.85 (0.76-0.91) | 0.84 (0.69-0.94) | 2.83 (0.27-11.00) | 0.89 (0.78-1.16) | 1.06 (0.82-1.60) |
| TGS-TB | 4 (212; 2224) | 0.81 (0.68-0.91) | 0.72 (0.48-0.90) | 0.66 (0.07-5.00) | 0.86 (0.70-1.12) | 0.90 (0.57-1.44) |

*(Continued)*

**Table 3.** (Continued)

| Tools | Studies (M/XDR; sizes) | Absolute Sensitivity (95% CI) | Absolute Specificity (95% CI) | Superiority Index | Relative Sensitivity (95% CI) | Relative Specificity (95% CI) |
|---|---|---|---|---|---|---|
| *KvarQ* | 4 (76; 198) | 0.80 (0.67-0.90) | 0.91 (0.72-0.99) | 4.52 (0.20-13.00) | 0.85 (0.69-1.11) | 1.14 (0.85-1.75) |
| *CASTB* | 4 (40; 71) | 0.69 (0.52-0.84) | 0.60 (0.34-0.85) | 0.09 (0.06-0.33) | **0.73 (0.52-0.98)** | 0.76 (0.40-1.33) |
| *MTBseq* | 1 (16; 502) | 0.79 (0.47-0.97) | 0.81 (0.48-0.97) | 2.78 (0.07-15.00) | 0.84 (0.48-1.17) | 1.01 (0.58-1.59) |
| *SAM-TB* | 2 (152; 928) | 0.87 (0.70-0.97) | 0.86 (0.60-0.98) | 5.96 (0.14-15.00) | 0.91 (0.72-1.19) | 1.07 (0.74-1.63) |
| *GenTB* | 1 (100; 110) | 0.96 (0.73-1.00) | 0.82 (0.52-0.97) | 8.30 (0.27-17.00) | 1.00 (1.00-1.00) | 1.00 (1.00-1.00) |

Abbreviations: TB, tuberculosis. M/XDR-TB, multidrug and extensively drug-resistant tuberculosis. CI, confidence interval.

Bold values indicate statistically significant differences.

relatively low sensitivity and specificity of *TBProfiler* (85% and 80%) could be attributed to the inherent limitations of pDST and the complex nature of resistance mechanisms, including 1) pDST is associated with interpretation (especially borderline resistance) and reproducibility issues; 2) pDST does not always correlate with genotypic resistance caused by mutations in *fabG1* and/or *ethA* genes [39,40]. For fluoroquinolone resistance, *TBProfiler* and other tools offer a comprehensive prediction for the entire fluoroquinolone class by aggregating phenotypes from moxifloxacin, ofloxacin, levofloxacin, and ciprofloxacin. Nevertheless, the effectiveness of these tools varies across different fluoroquinolones. *TBProfiler* appears to be the most effective tool for detecting levofloxacin resistance, whereas *Mykrobe* shows the highest accuracy for detecting moxifloxacin and ofloxacin resistance. This implies that the integration of multiple tools, or the updating of current algorithms and tools, is crucial for improving the overall accuracy and reliability.

Two key concepts are essential for the accuracy of bioinformatics tools. First, their diagnostic performance would benefit from amendments (especially algorithm and mutation) [41,42] and version updating. Since *KvarQ* is no longer updated, such a delay affects its performance. Additionally, the data suggested that in comparison to previous versions, an updated version of bioinformatics tools demonstrated improved predictive performance [6,7]. Second, the DST method is also a concern. Subgroup analyses stratified by the type of WHO recommendations for the pDST demonstrated that the pooled performance of bioinformatics tools is dependent on the pDST.

The continued development of WGS has facilitated the use of bioinformatics tools for TB drug resistance prediction. However, challenges remain in its application, such as inconsistent resistance calling between different tools, lack of a shared DST reference, rare mutations, and heteroresistance, as well as data ethics, privacy and security [43–46]. Therefore, establishing a standardized protocol and timely updated databases from a global perspective are needed to improve the performance of bioinformatics tools for TB management [47,48].

Although interesting findings were obtained by collating currently available data, several limitations should be acknowledged. First, due to limited data availability, several tools, such as *Resistance Sniffer* [49], *MycoVarP* [50], and *COMBAT-TB* [51], were not included in the analysis. Second, some bioinformatics tools are updated continuously, while others are not. Due to limited data, we could not take the version of bioinformatics tools into account in the data analyses, leading to a potential underestimation of test accuracy. Third, while *SAM-TB* demonstrated a high SI for certain drugs, caution should be exercised in interpreting these results, as the limited and high variability in data derived from only two studies resulted in wide confidence intervals and a high level of uncertainty in the estimate. Additionally, several tools were excluded from the ranking analysis due to their NMA results being based on data from only a single study.

**Table 4. Bayesian network meta-analysis of the diagnostic accuracy of bioinformatics tools for predicting resistance to any second-line anti-TB drugs.**

| Tools | Studies (M/XDR; sizes) | Absolute Sensitivity (95% CI) | Absolute Specificity (95% CI) | Superiority Index | Relative Sensitivity (95% CI) | Relative Specificity (95% CI) |
|---|---|---|---|---|---|---|
| Amikacin | | | | | | |
| TBProfiler | 14 (1203; 1582) | 0.88 (0.77-0.95) | 0.87 (0.74-0.95) | 2.49 (0.14-11.00) | 1.13 (0.81-2.54) | 1.08 (0.82-1.93) |
| Mykrobe | 9 (272; 645) | 0.75 (0.51-0.91) | 0.93 (0.80-0.99) | 2.55 (0.11-11.00) | 0.94 (0.59-2.05) | 1.16 (0.90-2.09) |
| PhyResSE | 5 (427; 555) | 0.77 (0.54-0.93) | 0.92 (0.76-1.00) | 3.02 (0.11-13.00) | 0.98 (0.59-2.16) | 1.14 (0.86-2.04) |
| TGS-TB | 3 (204; 291) | 0.74 (0.45-0.94) | 0.89 (0.64-1.00) | 2.03 (0.08-11.00) | 0.94 (0.50-2.15) | 1.11 (0.73-2.06) |
| KvarQ | 1 (52; 88) | 0.51 (0.00-1.00) | 0.90 (0.51-1.00) | 3.41 (0.07-17.00) | 0.66 (0.00-1.92) | 1.11 (0.61-1.91) |
| CASTB | 2 (24; 47) | 0.85 (0.36-1.00) | 0.63 (0.22-0.93) | 1.09 (0.06-9.00) | 1.08 (0.40-2.45) | 0.77 (0.26-1.48) |
| MTBseq | 1 (605; 677) | 0.81 (0.33-0.99) | 0.77 (0.28-0.99) | 1.90 (0.07-11.00) | 1.02 (0.39-2.29) | 0.95 (0.35-1.79) |
| SAM-TB | 2 (152; 198) | 0.85 (0.35-1.00) | 0.86 (0.53-0.99) | 4.30 (0.08-15.00) | 1.07 (0.44-2.35) | 1.06 (0.67-1.78) |
| GenTB | 1 (100; 110) | 0.86 (0.35-1.00) | 0.85 (0.44-1.00) | 4.74 (0.08-15.00) | 1.00 (1.00-1.00) | 1.00 (1.00-1.00) |
| Capreomycin | | | | | | |
| TBProfiler | 10 (826; 924) | 0.80 (0.56-0.95) | 0.82 (0.62-0.94) | 2.27 (0.11-9.00) | 9.55 (0.69-67.29) | 1.46 (0.68-3.69) |
| Mykrobe | 9 (272; 645) | 0.62 (0.39-0.84) | 0.92 (0.75-0.99) | 3.69 (0.11-11.00) | 6.45 (0.49-42.66) | 1.63 (0.85-4.11) |
| PhyResSE | 4 (220; 289) | 0.69 (0.38-0.97) | 0.85 (0.58-0.99) | 2.97 (0.09-11.00) | 7.00 (0.47-45.98) | 1.52 (0.68-3.99) |
| TGS-TB | 3 (204; 291) | 0.69 (0.29-0.98) | 0.85 (0.43-1.00) | 4.22 (0.08-15.00) | 4.68 (0.47-25.15) | 1.48 (0.56-3.60) |
| SAM-TB | 2 (152; 198) | 0.65 (0.29-0.97) | 0.83 (0.54-0.99) | 1.98 (0.08-11.00) | 6.23 (0.38-42.17) | 1.46 (0.64-3.65) |
| GenTB | 1 (100; 110) | 0.49 (0.21-0.85) | 0.81 (0.42-0.99) | 0.83 (0.07-7.00) | 4.02 (0.27-23.96) | 1.43 (0.47-3.61) |
| KvarQ | 1 (52; 88) | 0.61 (0.00-1.00) | 0.90 (0.61-1.00) | 5.57 (0.08-15.00) | 10.30 (0.00-78.50) | 1.59 (0.70-3.85) |
| CASTB | 1 (14; 37) | 0.57 (0.01-1.00) | 0.71 (0.23-1.00) | 2.95 (0.07-15.00) | 1.00 (1.00-1.00) | 1.00 (1.00-1.00) |
| Kanamycin | | | | | | |
| TBProfiler | 11 (1092; 1363) | 0.84 (0.68-0.95) | 0.83 (0.68-0.94) | 2.26 (0.14-9.00) | 1.79 (0.88-4.50) | 1.06 (0.79-1.88) |
| Mykrobe | 8 (262; 635) | 0.54 (0.26-0.84) | 0.87 (0.70-0.98) | 0.96 (0.08-7.00) | 1.11 (0.45-2.97) | 1.11 (0.84-1.97) |
| PhyResSE | 5 (427; 555) | 0.54 (0.28-0.79) | 0.86 (0.67-0.98) | 0.89 (0.08-5.00) | 1.13 (0.44-2.93) | 1.09 (0.78-1.86) |
| TGS-TB | 3 (204; 291) | 0.82 (0.48-0.99) | 0.86 (0.55-1.00) | 5.21 (0.09-17.00) | 1.75 (0.73-4.41) | 1.10 (0.64-1.96) |
| KvarQ | 1 (52; 88) | 0.53 (0.00-1.00) | 0.92 (0.62-1.00) | 4.79 (0.07-17.00) | 1.11 (0.00-3.60) | 1.17 (0.76-2.05) |
| CASTB | 1 (14; 37) | 0.68 (0.10-1.00) | 0.78 (0.25-1.00) | 3.73 (0.06-17.00) | 1.46 (0.17-4.28) | 1.00 (0.30-1.89) |
| MTBseq | 1 (605; 677) | 0.70 (0.21-0.99) | 0.72 (0.24-0.99) | 2.10 (0.06-15.00) | 1.50 (0.34-4.26) | 0.92 (0.30-1.72) |
| SAM-TB | 2 (152; 198) | 0.84 (0.39-1.00) | 0.88 (0.64-0.99) | 6.61 (0.14-17.00) | 1.78 (0.65-4.58) | 1.12 (0.81-1.85) |
| GenTB | 1 (100; 110) | 0.57 (0.18-0.94) | 0.82 (0.44-0.99) | 1.34 (0.07-11.00) | 1.00 (1.00-1.00) | 1.00 (1.00-1.00) |
| Levofloxacin | | | | | | |
| TBProfiler | 6 (370; 517) | 0.80 (0.57-0.93) | 0.70 (0.47-0.89) | 1.87 (0.11-9.00) | 1.84 (0.88-4.82) | 0.88 (0.54-1.72) |
| PhyResSE | 3 (273; 391) | 0.70 (0.43-0.91) | 0.72 (0.47-0.93) | 1.35 (0.09-9.00) | 1.53 (0.74-3.96) | 0.91 (0.55-1.77) |
| Mykrobe | 2 (66; 125) | 0.68 (0.30-0.91) | 0.75 (0.42-0.98) | 1.72 (0.09-9.00) | 1.48 (0.55-3.32) | 0.95 (0.51-1.86) |
| MTBseq | 1 (605; 677) | 0.83 (0.34-1.00) | 0.77 (0.31-0.99) | 5.85 (0.09-15.00) | 1.92 (0.51-4.85) | 0.98 (0.38-2.00) |
| KvarQ | 1 (52; 88) | 0.68 (0.21-0.93) | 0.87 (0.47-1.00) | 4.44 (0.09-13.00) | 1.52 (0.39-3.72) | 1.11 (0.57-2.17) |
| SAM-TB | 1 (52; 88) | 0.79 (0.32-0.98) | 0.68 (0.33-0.97) | 2.89 (0.09-11.00) | 1.77 (0.52-4.09) | 0.86 (0.40-1.77) |
| TGS-TB | 1 (14; 37) | 0.79 (0.36-0.99) | 0.55 (0.17-0.90) | 1.16 (0.08-9.00) | 1.74 (0.68-4.09) | 0.68 (0.24-1.35) |
| CASTB | 1 (14; 37) | 0.56 (0.17-0.89) | 0.84 (0.39-1.00) | 1.85 (0.08-11.00) | 1.00 (1.00-1.00) | 1.00 (1.00-1.00) |
| Moxifloxacin | | | | | | |
| TBProfiler | 9 (746; 1007) | 0.84 (0.65-0.94) | 0.82 (0.63-0.94) | 3.56 (0.14-11.00) | 1.76 (0.94-4.77) | 0.93 (0.71-1.32) |
| Mykrobe | 4 (86; 145) | 0.79 (0.55-0.95) | 0.85 (0.63-0.99) | 3.96 (0.11-13.00) | 1.64 (0.84-4.56) | 0.97 (0.71-1.41) |
| PhyResSE | 3 (273; 391) | 0.74 (0.52-0.91) | 0.81 (0.57-0.95) | 1.53 (0.09-9.00) | 1.55 (0.80-4.28) | 0.92 (0.64-1.34) |
| TGS-TB | 2 (150; 237) | 0.88 (0.60-1.00) | 0.74 (0.42-0.97) | 3.36 (0.09-13.00) | 1.83 (0.95-4.95) | 0.83 (0.51-1.25) |
| MTBseq | 1 (605; 677) | 0.68 (0.21-0.96) | 0.80 (0.32-1.00) | 3.30 (0.07-15.00) | 1.43 (0.35-4.00) | 0.91 (0.38-1.46) |
| KvarQ | 1 (52; 88) | 0.65 (0.15-0.97) | 0.84 (0.49-0.99) | 3.12 (0.08-13.00) | 1.36 (0.25-3.75) | 0.96 (0.56-1.40) |

*(Continued)*

**Table 4.** (Continued)

| Tools | Studies (M/XDR; sizes) | Absolute Sensitivity (95% CI) | Absolute Specificity (95% CI) | Superiority Index | Relative Sensitivity (95% CI) | Relative Specificity (95% CI) |
|---|---|---|---|---|---|---|
| *SAM-TB* | 1 (52; 88) | 0.65 (0.16-0.97) | 0.73 (0.35-0.95) | 1.22 (0.07-9.00) | 1.34 (0.30-3.88) | 0.83 (0.39-1.26) |
| *CASTB* | 1 (14; 37) | 0.58 (0.18-0.91) | 0.90 (0.58-1.00) | 2.63 (0.08-11.00) | 1.00 (1.00-1.00) | 1.00 (1.00-1.00) |
| Ofloxacin | | | | | | |
| *TBProfiler* | 9 (843; 1166) | 0.87 (0.73-0.94) | 0.93 (0.80-0.98) | 1.31 (0.14-5.00) | 2.06 (0.93-5.99) | 0.99 (0.85-1.38) |
| *Mykrobe* | 4 (24; 311) | 0.89 (0.67-0.99) | 0.93 (0.74-0.99) | 2.64 (0.14-7.00) | 2.15 (0.90-6.15) | 0.99 (0.80-1.38) |
| *PhyResSE* | 2 (221; 303) | 0.84 (0.59-0.95) | 0.93 (0.71-0.99) | 1.65 (0.14-7.00) | 1.99 (0.85-5.29) | 0.99 (0.76-1.40) |
| *TGS-TB* | 2 (150; 237) | 0.92 (0.70-1.00) | 0.91 (0.61-0.99) | 3.19 (0.14-9.00) | 2.14 (0.97-6.08) | 0.96 (0.64-1.32) |
| *CASTB* | 1 (14; 37) | 0.64 (0.15-0.93) | 0.95 (0.65-1.00) | 1.24 (0.11-7.00) | 1.00 (1.00-1.00) | 1.00 (1.00-1.00) |
| Ethionamide | | | | | | |
| *TBProfiler* | 11 (950; 1221) | 0.73 (0.56-0.87) | 0.67 (0.49-0.81) | 1.27 (0.20-5.00) | **7.38 (1.36-25.40)** | **0.71 (0.52-0.91)** |
| *PhyResSE* | 3 (275; 357) | 0.44 (0.13-0.76) | 0.79 (0.52-0.96) | 1.09 (0.14-5.00) | 4.33 (0.40-15.62) | 0.84 (0.56-1.10) |
| *TGS-TB* | 3 (204; 291) | 0.76 (0.46-0.96) | 0.69 (0.40-0.90) | 2.15 (0.33-7.00) | **7.63 (1.27-26.30)** | 0.73 (0.43-1.00) |
| *CASTB* | 1 (14; 37) | 0.20 (0.00-0.70) | 0.93 (0.59-1.00) | 1.59 (0.14-5.00) | 1.94 (0.00-10.26) | 0.99 (0.66-1.25) |
| *Mykrobe* | 3 (78; 101) | 0.18 (0.03-0.53) | 0.95 (0.74-1.00) | 1.57 (0.20-5.00) | 1.00 (1.00-1.00) | 1.00 (1.00-1.00) |
| Prothionamide | | | | | | |
| *TBProfiler* | 3 (187; 226) | 0.69 (0.33-0.94) | 0.76 (0.46-0.92) | 2.73 (0.20-7.00) | 1.20 (0.46-3.18) | 1.22 (0.67-2.55) |
| *Mykrobe* | 1 (100; 110) | 0.55 (0.14-0.90) | 0.78 (0.34-0.97) | 2.41 (0.20-7.00) | 0.94 (0.18-2.91) | 1.26 (0.54-2.79) |
| *GenTB* | 1 (100; 110) | 0.31 (0.00-0.87) | 0.68 (0.27-0.92) | 0.59 (0.14-3.00) | 0.56 (0.00-2.26) | 1.07 (0.45-2.25) |
| *SAM-TB* | 1 (100; 110) | 0.74 (0.21-1.00) | 0.68 (0.28-0.93) | 2.28 (0.14-7.00) | 1.00 (1.00-1.00) | 1.00 (1.00-1.00) |
| Para-aminosalicylic acid | | | | | | |
| *TBProfiler* | 10 (680; 912) | 0.62 (0.39-0.76) | 0.92 (0.77-0.98) | 0.29 (0.09-1.00) | 1.06 (0.56-2.78) | 1.15 (0.89-2.25) |
| *TGS-TB* | 3 (204; 291) | 0.92 (0.62-1.00) | 0.97 (0.82-1.00) | 4.03 (0.33-11.00) | 1.62 (0.82-4.46) | 1.22 (0.98-2.50) |
| *Mykrobe* | 3 (78; 101) | 0.91 (0.46-1.00) | 0.96 (0.80-1.00) | 4.67 (0.33-11.00) | 1.58 (0.74-4.28) | 1.21 (0.86-2.49) |
| *PhyResSE* | 2 (68; 91) | 0.82 (0.14-1.00) | 0.97 (0.81-1.00) | 4.50 (0.14-11.00) | 1.42 (0.25-3.93) | 1.22 (0.97-2.45) |
| *CASTB* | 1 (14; 37) | 0.55 (0.00-1.00) | 0.95 (0.65-1.00) | 2.49 (0.09-9.00) | 0.91 (0.00-3.22) | 1.18 (0.85-2.35) |
| *SAM-TB* | 1 (100; 110) | 0.71 (0.21-0.99) | 0.85 (0.38-0.99) | 0.54 (0.09-1.67) | 1.00 (1.00-1.00) | 1.00 (1.00-1.00) |

Abbreviations: TB, tuberculosis. M/XDR-TB, multidrug and extensively drug-resistant tuberculosis. CI, confidence interval.

Bold values indicate statistically significant differences.

Finally, the design of the included studies remains a concern, which could influence the reported accuracy. For example, DST was not performed for all aminoglycosides consistently, missing data were common across studies, and data were lacking, especially for second-line drugs and novel or repurposed drugs. Therefore, further investigations should be performed to validate our findings.

In conclusion, *TBProfiler*, *TGS-TB*, *Mykrobe*, *PhyResSE*, and *SAM-TB* are all efficient in predicting resistance to anti-TB drugs. Notably, *TBProfiler* has exceptional performance in predicting resistance to most anti-TB drugs, whereas *TGS-TB* excels in predicting resistance to pyrazinamide and certain second-line drugs. However, the reliability and effectiveness of *SAM-TB* warrants further investigation to be fully established. These varying rankings across different anti-TB drugs highlight the need for further optimization and improvement of bioinformatics tools. Given the diverse performances among tools and across drugs, selecting the most suitable prediction tool is crucial for tailoring treatment regimens. Bioinformatics tools and algorithms must be updated continuously, and novel resistance-associated mutations must be incorporated to maintain and improve their

efficacy. Ultimately, these findings will contribute to better detection and treatment of drug-resistant tuberculosis.

## Supporting information

**S1 Fig. Risk of bias and applicability concerns of included studies assessed using QUADAS-2.**
(PDF)

**S2 Fig. Forest plots of diagnostic yield for all bioinformatics tools across 14 anti-TB drugs.**
(PDF)

**S3 Fig. Summary receiver operating characteristic curves of all bioinformatics tools in predicting drug resistance to 14 anti-TB drugs.**
(PDF)

**S4 Fig. Publication bias of included studies assessed using Deek's funnel plot, analyzed separately for each drug.**
(PDF)

**S5 Fig. Network visualization of all included bioinformatics tools.**
(PDF)

**S1 File. PRISMA-NMA Checklist.**
(DOCX)

**S2 File. Electronic search strategy.**
(DOCX)

**S3 File. Workflow of the analytical process using streptomycin as an example, with a detailed description of the statistical methods and corresponding results.**
(PDF)

**S4 File. Detailed inclusion and exclusion criteria for 86 full-text articles.**
(XLSX)

**S1 Table. Characteristics of included studies.**
(DOCX)

**S2 Table. Original data for analysis.**
(DOCX)

**S3 Table. Comparison of features among bioinformatics tools.**
(DOCX)

**S4 Table. Methodological quality assessment of included studies based on QUADAS-2.**
(DOCX)

**S5 Table. Relative sensitivity and specificity of PhyResSE, Mykrobe, and TBProfiler stratified by culture-based pDSTs.**
(DOCX)

**S6 Table. Network meta-analysis of bioinformatics tools for predicting drug resistance to first- and second-line anti-TB drugs.**
(DOCX)

## Acknowledgments

We would like to express our gratitude to all the authors who have contributed to this systematic review and meta-analysis. Additionally, we acknowledge and appreciate the work of all authors whose papers were included in this study.

## Author contributions

**Conceptualization:** Ya-Li Chen, Mao-Shui Wang.

**Data curation:** Ya-Li Chen.

**Formal analysis:** Ya-Li Chen.

**Investigation:** Ya-Li Chen, Yu He, Cui-Ping Guan.

**Methodology:** Ya-Li Chen, Yu He, Cui-Ping Guan.

**Project administration:** Mao-Shui Wang.

**Writing – original draft:** Ya-Li Chen, Mao-Shui Wang.

**Writing – review & editing:** Victor Naestholt Dahl, Kan Yu, Yan-An Zhang.

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
