## [Decision Letter · Decision Letter 0]

28 Oct 2024

PGPH-D-24-02151

Diagnostic yield of eight user-friendly bioinformatics tools for predicting Mycobacterium tuberculosis drug resistance: a systematic review and network meta-analysis

Dear Dr. Chen,

Thank you for submitting your manuscript to PLOS Global Public Health. After careful consideration, we feel that it has merit but does not fully meet PLOS Global Public Health’s publication criteria as it currently stands. Therefore, we invite you to submit a revised version of the manuscript that addresses the points raised during the review process.

We look forward to receiving your revised manuscript.

Kind regards,

Kiatichai Faksri, Ph.D

Academic Editor

Journal Requirements:

1. As required by our policy on Data Availability, please ensure your manuscript or supplementary information includes the following: 

Additional Editor Comments (if provided):

Reviewers' comments:

Reviewer's Responses to Questions

**Comments to the Author**

1. Does this manuscript meet PLOS Global Public Health’s publication criteria ? Is the manuscript technically sound, and do the data support the conclusions? The manuscript must describe methodologically and ethically rigorous research with conclusions that are appropriately drawn based on the data presented.

Reviewer #1: Partly

Reviewer #2: Partly

2. Has the statistical analysis been performed appropriately and rigorously?

Reviewer #1: No

Reviewer #2: I don't know

3. Have the authors made all data underlying the findings in their manuscript fully available (please refer to the Data Availability Statement at the start of the manuscript PDF file)?

Reviewer #1: No

Reviewer #2: No

4. Is the manuscript presented in an intelligible fashion and written in standard English?

Reviewer #1: Yes

Reviewer #2: Yes

5. Review Comments to the Author

Reviewer #1: Authors has reviewed literature and compared various bioinformatics tools for predicting Mycobacterium tuberculosis drug resistance, with TBProfiler showing the best diagnostic accuracy for most anti-TB drugs. TGS-TB performed exceptionally for pyrazinamide and certain second-line drugs.

There are a few queries and concerns as listed below:

1. What are the criteria for for selecting the antibiotics list included in the study? We noticed that newer drugs like Bedaquiline, Delamanid, and Pretomanid, which are essential for treating rifampicin-resistant TB, were excluded from the analysis. Bedaquiline is a core drug in Rr-TB treatment regimens, and both Delamanid and Pretomanid have shown great potential in shortening MDR- and XDR-TB therapies and improving patient outcomes. Can author clarify the reasons for this exclusion?

Please refer: https://www.sciencedirect.com/science/article/pii/S1198743X2100464X#bib3 and https://www.ncbi.nlm.nih.gov/pmc/articles/PMC7825472/

1. Mutations in atpE gene, encoding subunit C of the ATP synthase, can prevent BDQ from binding to the C subunit, thus resulting in BDQ resistance https://www.ncbi.nlm.nih.gov/pmc/articles/PMC9976417/

2. synonymous mutation, T960C, in the fgd1 gene was uniformly found in DLM-resistant mtb https://www.ncbi.nlm.nih.gov/pmc/articles/PMC11170162/

3. Pretomanid and delamanid are pro-drugs that share the same activation pathway, the products of ddn, fgd1, fbiA-D. Loss-of-function and certain other mutations in any of these 6 genes have been associated with high delamanid/pretomanid resistance in M. tuberculosis https://www.ncbi.nlm.nih.gov/pmc/articles/PMC11069242/

2. How does the performance of other bioinformatics tools (such as TGS-TB, MTBseq, KvarQ, CASTB and SAM-TB missing in S3 Table and S3 Figure) compare in terms of sensitivity and specificity for predicting drug resistance in tuberculosis, particularly when benchmarked against WHO recommendations and phenotypic drug susceptibility testing (pDST)?

3. How were the bioinformatics tools or their performance selected for comparison in this study with respect to their Superiority Index (SI) in table 1, and what specific criteria/methods were used to determine their ranking and predictive accuracy for TB drug resistance?

4. In page no. 9, line no. 152, What factors contributed to the high risk of bias in sample and patient selection across the majority of studies, and how might the unblinded MTB resistance profiling impact the applicability of the findings? Additionally, what measures were taken to ensure low bias risk in the index and reference test domains, as well as in the flow and timing domain?

5. How did the Bayesian approach differ from traditional meta-analytic methods in assessing the diagnostic accuracy of the bioinformatics tools? In the abstract, the authors mention that the data were collected and pooled using random-effects meta-analysis and Bayesian network meta-analysis. However, the implications of the random-effects meta-analysis are not discussed in the manuscript as well as supplementary.

6. How were the heterogeneity (Differences between studies) and inconsistency (Differences in Results between tools) of studies accounted for in the Bayesian network meta-analysis? What implications do the results of the Bayesian network meta-analysis have for clinical practice and the management of TB drug resistance? As mentioned in the previous question, why did the author not discuss about random-effects meta-analysis anywhere in the manuscript, which is known to address heterogeneity and inconsistency?

7. In the study by Xiao et al., they used whole-genome sequencing (WGS) to predict drug resistance in Mycobacterium tuberculosis and found high accuracy for several antibiotics, including isoniazid and rifampicin. They mentioned that the Total Genotyping Solution for TB (TGS-TB) worked well for many drugs. In contrast, the review focused mainly on one antibiotic (pyrazinamide) and reported high effectiveness for it, along with good performance for other drugs like kanamycin and moxifloxacin. Why does the study show that TGS-TB is effective for many antibiotics, while the review only highlights one antibiotic? What could explain this difference in focus? (Please refer: https://www.nature.com/articles/s41598-023-29652-3).

8. Why did the author omit the ARIBA (Antimicrobial Resistance Identification By Assembly) tool, which demonstrates better specificity and lower major error compared to KvarQ, MTBseq, Mykrobe, and TB-profiler? Please see Table 5 in this publication for reference: https://www.ncbi.nlm.nih.gov/pmc/articles/PMC7004237/.

9. The raw data and codes were missing, and the methodology section was poorly written and lacked clarity in details. To ensure the reproducibility of the results in terms of statistical and scientific performance, it is essential for the authors to provide this information.

Reviewer #2: Introduction

This is a review of the article “Diagnostic yield of eight user-friendly bioinformatics tools for predicting Mycobacterium tuberculosis drug resistance: a systematic review and network meta-analysis” by Ya-Li Chen et al, submitted for publication to PLOS Global Public Health. The authors report an assessment of the diagnostic yield of 8 bioinformatics tools for Mycobacterium tuberculosis drug resistance prediction through a systematic review of published studies. Their comparison, based on several statistical methods, indicate TBProfiler is the best method overall but suggest other tools are better for the prediction of the resistance to certain drugs.

Merits

Although the authors were not the first to do a comparative assessment of bioinformatics tools for the prediction of Mycobacterium tuberculosis drug resistance (see for example reference 13 in the paper), they are responsible for the most comprehensive comparison, both in terms of the number of programs they evaluated, the search sources they employed and the amount of data they considered. Their search strategy, explained in Supplementary File 1, is simple yet effective and allowed the authors to build a comprehensive dataset. Although the data was already published, they were also the first to scrape the data and offer it in the form of written tables, which is a much-appreciated first step towards their availability in public databases.

The authors declared that this systematic review and network meta-analysis are compliant with suggested reporting guidelines. I found it to be compliant with most items in the PRISMA checklist [1]. I can also confirm that this protocol is registered in PROSPERO as stated in the text. The authors defined eligibility criteria that are specific and which I consider to be appropriate for the analysis. They also specifically report the risk of bias in their analysis which gives credibility to their protocol.

The methods are well laid out and the statistical analysis seems to be performed equally well, including in the case of subgroup analyses.

Critique

My main and very important concern with this work is that the data is not readily available and that the statistical analysis is not reproducible. I encourage the authors to publish both the data (if possible, or at least a protocol to extract the data from the original work) and the computational analysis pipeline in an open repository for others to check the results and confirm ain conclusions. The latter would be facilitative if the outdoors chose to conduct the statistical analysis in an open package like R instead of Stata. The openness is also needed given the advancement of bioinformatics tools, as this approach would allow anyone to extend the comparison and assess the performance as new methods become available.

Since the search strategy is relatively simple and crucial for the data collected, readers may want to know which other strategies the authors considered. Likewise, since the studies considered were independently evaluated by two reviewers, it is not transparent how they finally decided on any discrepancies.

Discussion

The authors have presented a very comprehensive analysis that promises to be useful for the entire community of researchers. However, I could not find the underlying data or statistical analysis needed to reproduce the findings, therefore I should raise caution in the credibility of this meta-analysis. I expect the authors to be capable of providing these and resubmitting the text for publication. If the analysis proves to be well-done, the conclusions presented by the authors will be supported by clear results. The manuscript is generally well written, with a correct use of English grammar and vocabulary.

Together with the comments in the previous sections, please note below some suggestions that may help to improve the manuscript:

I strongly suggest the authors include a warning around line 246, suggesting that using these bioinformatics tools without previous experience is problematic and might confound the researcher, no matter how well they perform and how friendly they are.

While SAM-TB performs best for the prediction of the performance of some drugs in the Bayesian network meta-analysis, it is not considered in the resulting comparative analysis. The authors should explain better and earlier in the manuscript why they decided to leave this program out. Moreover, they should figure out if they still want to include this method or not; perhaps they require an additional criterium that would take program out of the analysis.

Given the activity in the field, it is not surprising to find 19 more search results in PubMed only as of October 25, 2024, using the authors' search strategy. Therefore, I would recommend the authors to update their results and, provided the papers and the data satisfy the eligibility criteria, incorporate the extra data in their analysis, which should be straightforward if the analysis is reproducible

Although the report is largely compliant with the PRISMA-NMA checklist [1], I suggest the authors particularly check if the abstract includes funding information and registration number.

Introduction. In line 62, "(...) drug resistance (i.e., isoniazid (...)”, I suggest replacing 'i.e.' with 'e.g.' as not all first-line drugs are mentioned there.

Figure 2. I think the data would be better presented as a table to encourage others to extract it. Regardless, the title and data in the columns need a better alignment. Also, it would be useful to explain again in this and other figure's captions what is the meaning of abbreviations like "M/XDR-TB".

Supplementary Figure 3. For clear identification, if possible, name the method and antibiotic used at the top of each panel instead of in the caption

Table 1. For a clearer analysis, please separate first- and second-order drugs with a horizontal line or in two separate tables. Also, please indicate in the caption why are some values stressed in bold.

Results > Bayesian network meta-analysis. Lines 202 to 205 need reviewing, because for example, the pooled sensitivity and specificity of the bioinformatics tools are also listed in Table 1.

Results > Bayesian network meta-analysis. The Bayesian network meta-analysis described between lines 207 to 213 seems to be missing the resistance prediction to levofloxacin. I suggest the authors add a the analysis here.

Results > Bayesian network meta-analysis. Add 'respectively' nearly at the end of line 210.

Bibliography

https://www.prisma-statement.org/nma

6. PLOS authors have the option to publish the peer review history of their article (what does this mean? ). If published, this will include your full peer review and any attached files.

**Do you want your identity to be public for this peer review?** For information about this choice, including consent withdrawal, please see our Privacy Policy .

Reviewer #1: **Yes: ** Anshu Bhardwaj

Reviewer #2: **Yes: ** Nicolas Palopoli

---

## [Decision Letter · Decision Letter 1]

17 Mar 2025

Diagnostic yield of nine user-friendly bioinformatics tools for predicting Mycobacterium tuberculosis drug resistance: a systematic review and network meta-analysis

PGPH-D-24-02151R1

Dear Chen,

We are pleased to inform you that your manuscript 'Diagnostic yield of nine user-friendly bioinformatics tools for predicting Mycobacterium tuberculosis drug resistance: a systematic review and network meta-analysis' has been provisionally accepted for publication in PLOS Global Public Health.

Best regards,

Kiatichai Faksri, Ph.D

Academic Editor

Dear Authors,

We are pleased to inform you that after reviewing the revised version of your manuscript, two reviewers have recommended its acceptance. Based on their evaluations, we have decided to accept your work for publication in PLOS Global Public Health.

However, we kindly ask you to address one remaining concern raised by a reviewer regarding the inclusion of the complete code. Additionally, please incorporate the discussion points from the reviewer’s comments into the main text.

We appreciate your efforts in improving the manuscript and look forward to receiving the final version with the requested revisions.

Best regards,

Kiatichai Faksri

Associate Editor

Reviewer Comments (if any, and for reference):

Reviewer's Responses to Questions

**Comments to the Author**

1. If the authors have adequately addressed your comments raised in a previous round of review and you feel that this manuscript is now acceptable for publication, you may indicate that here to bypass the “Comments to the Author” section, enter your conflict of interest statement in the “Confidential to Editor” section, and submit your "Accept" recommendation.

Reviewer #1: All comments have been addressed

Reviewer #2: All comments have been addressed

2. Does this manuscript meet PLOS Global Public Health’s publication criteria ? Is the manuscript technically sound, and do the data support the conclusions? The manuscript must describe methodologically and ethically rigorous research with conclusions that are appropriately drawn based on the data presented.

Reviewer #1: Partly

Reviewer #2: Yes

3. Has the statistical analysis been performed appropriately and rigorously?

Reviewer #1: I don't know

Reviewer #2: I don't know

4. Have the authors made all data underlying the findings in their manuscript fully available (please refer to the Data Availability Statement at the start of the manuscript PDF file)?

Reviewer #1: Yes

Reviewer #2: Yes

5. Is the manuscript presented in an intelligible fashion and written in standard English?

Reviewer #1: Yes

Reviewer #2: Yes

6. Review Comments to the Author

Reviewer #1: Thanks for addressing most comments related to the manuscript. A strong recommendation is to include most part of the discussion during the review process in the main manuscript for the benefit of the readers in clearly defining the scope of the comparisons made in the study.

Also, kindly provide all code in a manner which makes it possible for others to reproduce the study.

Reviewer #2: This is my review of the revised version of Manuscript Number PGPH-D-24-02151-R1, titled "Diagnostic yield of nine user-friendly bioinformatics tools for predicting Mycobacterium tuberculosis drug resistance: a systematic review and network meta-analysis",

I would like to thank the authors for considering all my previous comments and addressing them in their work and the revised version of the article. The inclusion of new data and files has addressed many of the issues I raised in my previous review, and I particularly appreciate the inclusion of Supplementary File 4.

Although it still lacks details and data to be fully reproducible, I acknowledge this version of the article is a step forward toward reproducibility and open science. The authors now provide an extended and thoroughly revised Methods section. The careful protocol focused on streptomycin is illustrative of the general protocol. The authors justify the use of Stata for data analysis. Although I do not agree with the authors, and I don't consider "the extensive work of our analyses" a valid reason for not enabling the replication of analyses, I consider that the information provided could suffice in this case. I see that the inclusion/exclusion criteria are not sufficiently explained, but I understand that some decisions rely on the experience and careful curations of the literature by the authors and the availability of tables where the authors report which articles have been considered makes it possible to replicate that selection step.

Overall I congratulate the others for devising a much-improved version of the article that the Editor may now consider suitable for publication in PLOS Global Public Health.

7. PLOS authors have the option to publish the peer review history of their article (what does this mean? ). If published, this will include your full peer review and any attached files.

**Do you want your identity to be public for this peer review?** For information about this choice, including consent withdrawal, please see our Privacy Policy .

Reviewer #1: No

Reviewer #2: **Yes: ** Nicolas Palopoli
